# Efficient Dilated Squeeze and Excitation Neural Operator for Differential Equations

**Prajwal Chauhan**                                    *pc3377@nyu.edu*
*Engineering Division*
*New York University Abu Dhabi*
*Abu Dhabi, UAE*

**Salah Eddine Choutri**                               *sc8101@nyu.edu*
*NYUAD Research Institute*
*New York University Abu Dhabi*
*Abu Dhabi, UAE*

**Saif Eddin Jabari**                                  *sej7@nyu.edu*
*Engineering Division*
*New York University Abu Dhabi*
*Abu Dhabi, UAE*

**Reviewed on OpenReview:** *https://openreview.net/forum?id=Xl942THEUa*

## Abstract

Fast and accurate surrogates for physics-driven partial differential equations (PDEs) are essential in fields such as aerodynamics, porous media design, and flow control. However, many transformer-based models and existing neural operators remain parameter-heavy, resulting in costly training and sluggish deployment. We propose D-SENO (Dilated Squeeze-Excitation Neural Operator), a lightweight operator learning framework for efficiently solving a wide range of PDEs, including airfoil potential flow, Darcy flow in porous media, pipe Poiseuille flow, and incompressible Navier–Stokes vortical fields. D-SENO combines dilated convolution (DC) blocks with squeeze-and-excitation (SE) modules to jointly capture wide receptive fields and dynamics alongside channel-wise attention, enabling both accurate and efficient PDE inference. Carefully chosen dilation rates allow the receptive field to focus on critical regions, effectively modeling long-range physical dependencies. Meanwhile, the SE modules adaptively recalibrate feature channels to emphasize dynamically relevant scales. Our model achieves training speed of up to $\approx 20\times$ faster than standard transformer-based models and neural operators, while also surpassing (or matching) them in accuracy across multiple PDE benchmarks. Ablation studies show that removing the SE modules leads to a slight drop in performance. Code: github.com/pj1911

## 1 Introduction

Partial differential equations (PDEs) dictate many natural phenomena, from synoptic-scale weather systems to wingtip vortices, to subsurface multiphase flow, and many more. Conventional numerical solvers such as finite difference, finite volume, spectral, and finite element schemes deliver high fidelity, yet their cost explodes with mesh resolution, geometric complexity, and parametric sweeps Quarteroni & Valli (2008). To close this computational gap, the focus is increasingly shifting to machine learning surrogates, which reduce the cost of high-resolution data in training and then provide accurate predictions at lower cost.

Recent neural PDE solvers range from physics-informed regressor to fully data-driven operator learners, each occupying a different point on the spectrum of accuracy, flexibility, and computational efficiency. At the physics-heavy end of the spectrum, physics-informed neural networks (PINNs) embed the governing equations, initial conditions, and boundary conditions directly into the loss via automatic differentiation, allowing them to train on sparse or even entirely unlabeled solution data. However, their stiff multiscale residuals make optimization delicate, and each change in geometry or forcing profile usually requires an expensive full retraining or at least substantial fine-tuning Raissi et al. (2019). To reduce training and inference costs, recent work has focused on operator learning, where a single neural network approximates the solution operator, mapping input functions to their corresponding PDE solutions across a family of problems.

The first operator–learning model, DeepONet Lu et al. (2019), represents the mapping with two sub-networks: a branch net that processes the input function and a trunk net that evaluates basis functions at the query points. Its mesh-free design and small parameter count make it attractive for parametric problems, yet running both nets at every spatial point slows training and inference as the grid becomes large Lu et al. (2021). To mitigate this scaling bottleneck and capture interactions that extend across the entire domain, subsequent work turned to architectures with built-in global couplings.

The Fourier Neural Operator (FNO) addresses these goals by exchanging, to certain extent, spatial locality for spectral mixing Li et al. (2020): Fast Fourier transforms (FFT) reduce complexity to $\mathcal{O}(N \log N)$ and grant mesh invariance, but the resulting all-to-all convolution can blur sharp discontinuities and inflate GPU memory on large three-dimensional domains such as wind-tunnel flows. To enhance spatial locality, some variants perform FFTs over overlapping windows or decompose the integral kernels onto wavelet bases Gupta et al. (2021), Tripura & Chakraborty (2023). While, these approaches partially recover local interactions, the necessary window overlap or boundary padding adds computational overhead and can introduce artifacts that complicates training.

In fixed Cartesian grids, U-NO Rahman et al. (2022) combines global Fourier-based integral layers with a U-Net-style encoder–decoder and multiscale skip connections. whereas F-FNO Tran et al. (2021) employs separable spectral (Fourier) layers factorized across dimensions alongside residual shortcuts. Like FNO, both U-NO and F-FNO inherit its core limitations: they depend on global FFTs, making them naturally suitable for periodic and uniform grids. They also share FNO's high memory and computational costs, especially in higher dimensions. Their spectral kernels require deeper architectures or windowed variants to capture fine-scale locality, which further increases overhead and can destabilize training.

Transformer-based neural operators have recently advanced the state of the art by leveraging multi-head attention to capture long-range dependencies. However, standard self-attention scales quadratically with the number of spatial grid points $\mathcal{O}(N^2)$, making high-resolution problems computationally demanding without aggressive sparsification or patching strategies Cao (2021); Guibas et al. (2021).

Among these models, Transolver Wu et al. (2024) introduces a physics-aware clustering mechanism that groups flow-aligned regions before applying intra- and inter-cluster attention. This approach yields state-of-the-art accuracy across several PDE benchmarks, outperforming FNO, UNO, and prior transformer-based baselines. Despite its accuracy gains, Transolver incurs substantial computational overhead: its iterative clustering, multi-stage refinement, and large hidden dimensions lead to training and inference times an order of magnitude higher than many neural operator architectures.

**Our Contribution.** We introduce D-SENO, a lightweight neural operator architecture designed for fast and accurate surrogate modeling of partial differential equations (PDEs). While previous convolution-based neural operators (e.g., CNO Raonić et al. (2023)) rely on fixed receptive fields and uniform filter patterns, D-SENO advances this paradigm by effectively integrating two key components within a unified framework:

1. **Dilated Convolutional Blocks:** We apply residual blocks with non-uniform, dataset-specific dilation rates across spatial dimensions. This allows the model to flexibly expand its receptive field and capture multiscale spatial dependencies, while maintaining strict locality and linear computational complexity. In contrast to the Dilated Convolution Neural Operator (DCNO) Xu et al. (2025), which employs all dilation rates in every process block, our approach customizes and selects a single,

dataset-specific rate per block and connects them via residual links for improved expressiveness and adaptability.

2. **Convolutional Channel Attention:** To enhance channel-wise expressiveness without global operations, we incorporate Squeeze-and-Excitation (SE) blocks Hu et al. (2018) using pointwise convolutions. These blocks dynamically recalibrate feature channels based on global context, improving relevance to physical patterns without incurring the overhead of transformer-style attention.

This architecture remains strictly local and fully convolutional, requiring neither Fourier transforms nor self-attention mechanisms. All layers preserve spatial resolution and scale linearly with input size, making D-SENO particularly efficient for high-resolution PDE inference.

We validate our model on diverse PDE benchmarks, including airfoil potential flow, Poiseuille pipe flow, heterogeneous Darcy flow, and Navier–Stokes vortices. D-SENO achieves high accuracy while offering up to a 20× speedup in training and inference over state-of-the-art neural operator and transformer-based surrogates.

The remainder of this paper is organized as follows: we begin by reviewing the relevant prior work, followed by discussing the core components of our approach, then detail the architecture and training protocol of D-SENO, and finally evaluate its accuracy, efficiency, and generalization across multiple PDE benchmarks.

## 2 Related Work

While the literature contains a wide array of neural PDE surrogates, we focus our discussion on representative models that are most relevant to our approach: the Fourier Neural Operator (FNO), the Convolutional Neural Operator (CNO), and the Dilated Convolutional Neural Operator (DCNO). We briefly summarize their core methodologies and highlight their limitations in the context of our work.

**Fourier Neural Operator (FNO).** The Fourier Neural Operator (FNO) Li et al. (2020) was among the first neural operators to approximate mappings between function spaces using the Fast Fourier Transform (FFT). Instead of learning convolutional kernels in physical space, FNO projects the input function onto a spectral basis via FFT, applies learned complex-valued weights to a subset of low-frequency modes, and then returns to the physical domain using an inverse FFT. This produces a global convolution-like behavior over the spatial domain.

At each layer, the update rule combines a spectral convolution (in frequency space) with a pointwise convolution, enabling the model to capture long-range dependencies. Since FFTs operate globally, FNO exhibits resolution invariance and has shown strong performance on structured and periodic PDE problems. However, this global nature introduces limitations. FNO may struggle with problems involving localized features (e.g., sharp shocks), complex or irregular boundary conditions, or uneven spatial behavior due to the lack of spatial localization in spectral kernels Chauhan et al. (2025).

In contrast, our method replaces global spectral operations with spatially localized dilated convolutions, allowing receptive fields to grow in a controlled manner. Combined with channel-wise recalibration via Squeeze-and-Excitation (SE) blocks, this leads to a lightweight architecture better suited for capturing localized physical patterns.

**Convolutional Neural Operator.** The Convolutional Neural Operator (CNO) Raoníc et al. (2023) is a fully convolutional architecture for learning resolution-specific mappings arising in PDEs, designed to be robust across varying discretizations and noise levels. It builds on standard convolutional layers arranged in an encoder–decoder (U-Net-like) fashion, with residual blocks and anti-aliased downsampling to maintain spatial fidelity, while avoiding the use of global Fourier or attention-based mechanisms. The model emphasizes locality and generalization, but it does not incorporate direction-dependent non-uniform convolutions for data-dependent receptive field shaping, nor explicit input-adaptive feature mixing through channel-wise recalibration mechanisms. Our approach addresses these limitations by integrating non-uniform dilated

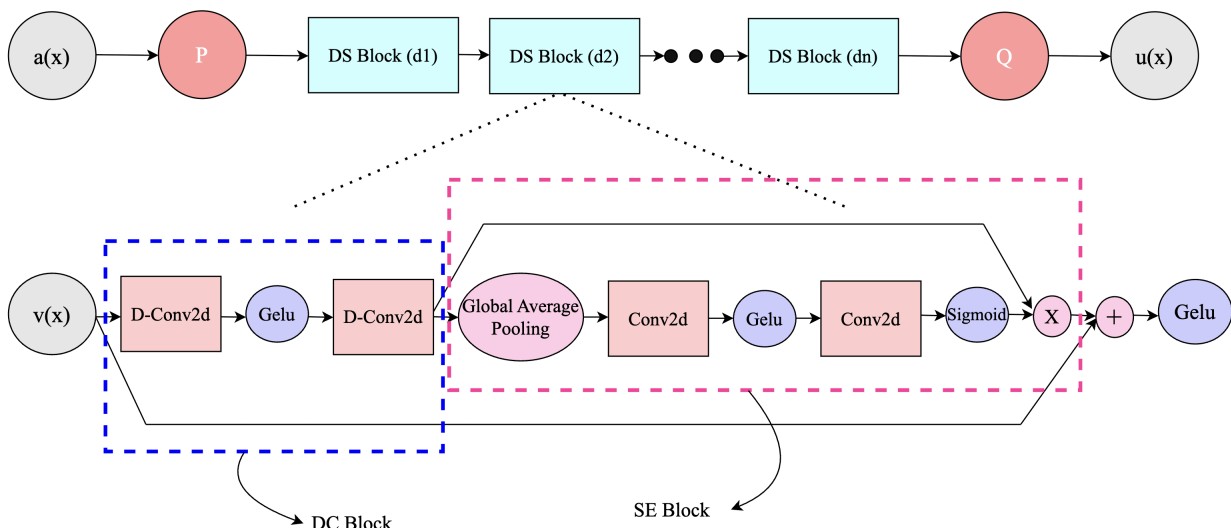

Figure 1: D-SENO architecture

convolutions for multiscale spatial context and Squeeze-and-Excitation blocks for input-aware channel recalibration, yielding improved adaptability in both space and feature selection.

**Dilated Convolution Neural Operator.** The Dilated Convolution Neural Operator (DCNO) Xu et al. (2025) introduces a hybrid operator learning architecture that combines spectral and spatial mechanisms. Specifically, it inserts dilated convolutional blocks between Fourier layers, aiming to capture both high-frequency local details and low-frequency global behavior. The use of dilated convolutions allows for an expanded receptive field without additional parameters, while the Fourier layers provide resolution-invariant global context. Although DCNO leverages FNO components to preserve operator generalization across discretizations, the added convolutional modules improve efficiency and local feature representation. While this design improves expressiveness, it also introduces several limitations. First, the reliance on FFT makes DCNO best suited for periodic domains and structured grids, limiting its applicability in problems with irregular or non-periodic boundaries. Second, although dilated convolutions are lightweight, the interleaved Fourier layers increase model's complexity and reduces it's flexibility. Third, the fixed dilation schedule may not generalize optimally across different datasets or spatial scales. These trade-offs affect both runtime performance and adaptability across diverse PDE regimes.

## 3 Background

Deep neural networks have demonstrated remarkable effectiveness in modeling spatially structured data across a wide range of domains. To handle complex patterns that vary across spatial scales, modern architectures often incorporate mechanisms that expand receptive fields and adaptively recalibrate internal representations. This section reviews two such components: Dilated Convolutions (DC) and Squeeze-and-Excitation (SE) network blocks.

**Dilated Convolutions.** Dilated convolutions, also referred to as atrous convolutions, are a variant of the standard convolutional operation that introduces a spacing factor between kernel elements. This modification enables the filter to cover a larger receptive field without increasing the number of parameters or reducing resolution. Originally explored in wavelet analysis Holschneider et al. (1987); Shensa (1992), dilated convolutions were later adopted in deep learning for tasks such as semantic segmentation Yu & Koltun (2015), and more recently, operator learning Xu et al. (2025).

Let $f : \mathbb{Z}^2 \to \mathbb{R}$ be a discrete input function and $k : \Omega_r \to \mathbb{R}$ a convolutional kernel with support size $(2r+1)^2$, $r \in \mathbb{Z}_{\geq 0}$. The conventional 2D convolution is given by:

$$(f * k)(x) = \sum_{s+t=x} f(s) \cdot k(t). \tag{1}$$

A dilated convolution introduces a dilation factor $l \in \mathbb{N}$. We denote the $l$-dilated convolution by $*_l$, defined as:

$$(f *_l k)(x) = \sum_{s+l \cdot t=x} f(s) \cdot k(t). \tag{2}$$

This effectively stretches the kernel by inserting $l-1$ zeros between adjacent filter elements along each axis. When $l = 1$, the operation reduces to the standard convolution. Higher dilation rates allow the network to aggregate information across a wider area of the input without pooling or strided operations. In our case, we allow different dilation factors along each spatial axis, denoted by $l_x, l_y \in \mathbb{N}$, yielding a direction-dependent dilated convolution $*_{(l_x, l_y)}$. This results in non-uniform receptive field expansion along the spatial dimensions, enabling the operator to capture elongated and orientation-specific structures by aggregating spatial context at different rates along the horizontal and vertical directions while preserving resolution.

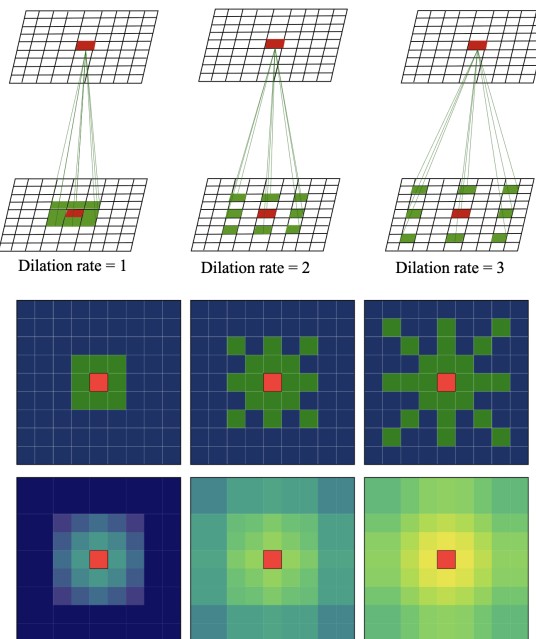

Figure 2: **Top.** Dilated Convolution for 3 dilation rates, **Middle.** Cumulative tap locations for dilation rates $1, 2, 3$ (left to right), **Bottom** Receptive field growth with propagation for dilation rates $1, 2, 3$, each applied twice (left to right)

**Squeeze-and-Excitation Blocks.** To enhance the representational capacity of convolutional neural networks, Hu et al. (2018) introduced the Squeeze-and-Excitation (SE) block, a lightweight architectural unit that adaptively recalibrates channel-wise feature responses. The core idea is to model inter-channel dependencies explicitly, allowing the network to emphasize informative features and suppress less useful ones based on global context.

The SE block acts on the feature maps $\mathbf{U}$, generated by a preceding transformation. Let $\mathbf{U} \in \mathbb{R}^{H \times W \times C}$ be the output of the DC block, the SE block then recalibrates these features in two stages:

**Squeeze.** In the first stage, global information is captured by compressing each channel-wise feature map $\mathbf{u}_c \in \mathbb{R}^{H \times W}$ (for channel index $c$) into a single scalar descriptor via global average pooling:

$$z_c = \frac{1}{H \times W} \sum_{i=1}^{H} \sum_{j=1}^{W} \mathbf{u}_c(i, j), \tag{3}$$

resulting in a vector $\mathbf{z}$ that summarizes the global statistics for each channel. In implementation, this vector is typically stored as a $1 \times 1 \times C$ tensor, so that it can be broadcast back over the feature maps, but in the following we identify it with its flattened vector form $\mathbf{z} \in \mathbb{R}^C$ for notational convenience.

**Excitation.** In the second stage, the aggregated descriptor $\mathbf{z}$ is then passed through a small channel-wise gating mechanism. Although implemented using two successive $1 \times 1$ convolutions in our architecture, this operation is equivalent to applying two fully connected layers to the flattened descriptor and is expressed in matrix form for clarity:

$$\hat{\mathbf{z}} = \phi\left(W_1 \mathbf{z} + \mathbf{b}_1\right), \tag{4}$$

$$\mathbf{s} = \sigma\left(W_2 \hat{\mathbf{z}} + \mathbf{b}_2\right), \tag{5}$$

where $W_1 \in \mathbb{R}^{\frac{C}{r} \times C}$ and $W_2 \in \mathbb{R}^{C \times \frac{C}{r}}$ are learnable weight matrices, $\mathbf{b}_1 \in \mathbb{R}^{\frac{C}{r}}$ and $\mathbf{b}_2 \in \mathbb{R}^C$ are bias terms, $\phi$ is the GELU activation function, and $\sigma$ is the element-wise sigmoid function. The reduction ratio $r \in \mathbb{Z}_+$ (with $r = 1$ meaning no reduction) explicitly controls the width of the intermediate channel representation: the first layer reduces the channel dimension from $C$ to $C/r$, and the second layer restores it from $C/r$ back to $C$. The output $\mathbf{s} \in \mathbb{R}^C$ provides a set of channel-wise modulation weights.

Finally, the original feature maps are reweighted channel-wise using the learned modulation weights. The vector $\mathbf{s} \in \mathbb{R}^C$, in practice, is expanded to $\mathbb{R}^{1 \times 1 \times C}$ and broadcast over spatial locations, and each channel slice $\mathbf{u}_c \in \mathbb{R}^{H \times W}$ of $\mathbf{U} \in \mathbb{R}^{H \times W \times C}$ is scaled by its corresponding weight $s_c$:

$$\tilde{\mathbf{u}}_c(i, j) = s_c \cdot \mathbf{u}_c(i, j), \tag{6}$$

yielding a recalibrated feature map $\tilde{\mathbf{U}} \in \mathbb{R}^{H \times W \times C}$ in which each channel's contribution is adjusted based on global context.

SE blocks have been successfully integrated into various backbone architectures (e.g., ResNet, Inception) and demonstrate consistent improvements in classification accuracy at a low computational cost Hu et al. (2018).

## 4 Method

### 4.1 Learning solution operator for PDEs

A (solution) operator $\mathcal{G} : \mathcal{A} \to \mathcal{U}$ maps elements from an input function space $\mathcal{A}$ (e.g., boundary conditions, source terms, or coefficients of a PDE) to an output function space $\mathcal{U}$, which typically contains the corresponding solutions of the PDE. The precise definition of these spaces depends on the particular PDE under consideration and the physical domain of interest.

Solving a PDE can thus be interpreted as evaluating $u = \mathcal{G}(a)$, where $a \in \mathcal{A}$ is the input function and $u \in \mathcal{U}$ is the resulting solution field. A neural operator aims to approximate this mapping $\mathcal{G}$ in a data-driven manner by learning a parametric representation $\mathcal{G}_\theta \approx \mathcal{G}$, which generalizes to new inputs $a \in \mathcal{A}$ that were not seen during training.

In the following subsection, we introduce our proposed model, D-SENO, which is specifically designed to learn such solution operators for PDEs while efficiently capturing both local and global dependencies through dilated convolutions and squeeze-and-excitation mechanism.

## 4.2 D-SENO

We develop a fully convolutional architecture that combines multiscale dilated convolutions with SE-based adaptive channel-wise recalibration. The model follows a lift-process-project structure: the input field $a$ is first lifted to a latent space, processed by a sequence of Dilated-Squeeze (DS) layers, and then projected to the output solution $u$. An overview is shown in Figure 1.

Each DS layer, denoted as $\mathcal{B}_i$, begins with a dilated convolution block (DC) which consists of two stacked $k \times k$ convolution layers (typically $3 \times 3$) that use dataset-specific dilation rates applied independently along the $x$- and $y$-axes. These dilation rates (for details, we refer the reader to supplementary materials) allow the receptive field to grow flexibly without down sampling or increasing the number of parameters. Following the dilated convolutions, a lightweight SE block adjusts the importance of each feature channel. We modify the original squeeze-and-excitation block by retaining global average pooling but substituting the fully-connected layers with two $1 \times 1$ convolutions, separated by a GELU activation. These layers reduce and then restore the number of channels, controlled by a reduction factor $r \in \mathbb{Z}_+$ as shown in section 3. Global average pooling is used to compute channel-wise statistics, which are transformed into scaling weights and applied via element wise multiplication. A residual connection adds the DS block input to the recalibrated output, followed by a GELU activation. This allows each $\mathcal{B}_i$ to modulate both spatial and channel information, improving expressiveness and efficiency.

The full model is composed of three stages:

1. **Lifting:** A pointwise convolutional layer $P$ transforms the input $a$ into a high-dimensional latent space.

2. **Processing:** A sequence of $n$ DS blocks $\mathcal{B}_1, \ldots, \mathcal{B}_n$, each preserving spatial resolution and expanding receptive fields.

3. **Projection:** A final pointwise layer $Q$ maps the processed features to the output $u$.

All spatial operations preserve resolution; no striding or downsampling is used. Global average pooling occurs only within SE blocks to compute channel-wise weights, without altering spatial dimensions.

Formally, the model approximates the solution operator $\mathcal{G}$ by a learned mapping $\mathcal{G}_\theta : \mathcal{A} \to \mathcal{U}$, producing an approximate solution function:

$$\hat{u}(\cdot) = \mathcal{G}_\theta(a)(\cdot) = Q \circ (\mathcal{B}_n \circ \cdots \circ \mathcal{B}_1) \circ P(a)(\cdot), \tag{7}$$

where $\theta$ denotes the model parameters.

To assess the impact of dilation and recalibration, we conduct ablation studies and refer the reader to the supplementary materials for details on the configurations evaluated.

## 5 Experiments

We evaluate the proposed D-SENO on the PDE benchmarks used by previous operator learners and transformer based models. The suite spans structured and regular grids, mirroring the experimental protocol of Transolver Wu et al. (2024) for direct comparability.

**Benchmarks.** Table 1 presents four benchmark PDE problems: (1) airfoil potential flow, (2) Poiseuille pipe flow, (3) heterogeneous Darcy flow (these three are steady), and (4) the time-dependent incompressible Navier–Stokes equations (unsteady), brief per-dataset summaries are given below.

### 5.1 Airfoil

The airfoil dataset comprises transonic Euler simulations of inviscid flow around randomly deformed NACA–0012 airfoils. The dataset (2D) consists of 1,000 training and 200 test samples of airfoil geometries, each

Table 1: Summary of experiment benchmarks with dimensions and mesh size.

| BENCHMARK | #DIM | MESH |
|---|---|---|
| Airfoil | 2D | $221 \times 51$ |
| Pipe | 2D | $129 \times 129$ |
| Darcy | 2D | $85 \times 85$ |
| Navier–Stokes (time) | $2D + T$ | $64 \times 64 \times 20$ |

discretized on a structured mesh of size $221 \times 51$. For each case, the input tensor has shape $221 \times 51 \times 2$, encoding the airfoil's structural geometry at every mesh point, and the target output is the local Mach number (the ratio of the local flow velocity to the speed of sound) at each point of shape $221 \times 51 \times 1$. All shapes are generated by deforming the baseline NACA-0012 profile provided by the National Advisory Committee for Aeronautics Li et al. (2023b).

## 5.2 Pipe

The pipe-flow dataset contains simulations of incompressible, viscous fluid moving through two-dimensional pipes whose centerlines are randomly curved. The dataset (2D) comprises of 1,000 training and 200 test samples of pipe geometries, each discretized on a structured mesh of size $129 \times 129$. For each case, the input tensor has shape $129 \times 129 \times 2$, which encodes the structural geometry of the pipe at each mesh point, and the target output is the horizontal velocity of the fluid at each point of shape $129 \times 129 \times 1$. All samples are generated by varying the pipe's centerline geometry Li et al. (2023b).

## 5.3 Darcy

Darcy flow describes the steady, viscous motion of an incompressible fluid through a porous medium. The dataset (2D) consists of 1,000 training and 200 testing samples of two-dimensional porous media, originally defined on a $421 \times 421$ regular grid and down-sampled to $85 \times 85$ for experiments. Each input tensor of shape $85 \times 85 \times 1$ encodes the binary structure of the porous medium, and the target output of shape $85 \times 85 \times 1$ gives the fluid pressure at each grid point. Different cases contain different medium structures Li et al. (2020).

## 5.4 Navier Stokes

The Navier–Stokes dataset models incompressible, viscous flow on a unit torus with constant density and viscosity $10^{-5}$ Li et al. (2020). Each trajectory is discretized on a $64 \times 64$ regular grid. The input tensor of shape $64 \times 64 \times 2 \times 10$ contains the two velocity components over the past 10 time steps, and the target output of the same shape predicts the velocity field for the next 10 steps. A total of 1,000 simulations with varying initial conditions are used for training, and 200 new trajectories are reserved for testing.

**Baselines and Implementation Protocol.** We compare D-SENO with more than ten representative models: neural operators (FNO Li et al. (2020), U-NO Rahman et al. (2022), LSM Wu et al. (2023)), transformer-style PDE solvers (GNOT Hao et al. (2023), Factorized FNO Tran et al. (2021)). Transolver Wu et al. (2024) represents the previous state-of-the-art on all benchmarks. Moreover, we observe that modest tweaks to FNO such as replacing `ReLU` with `GELU` activations, removing batch normalization, preserving sufficient Fourier modes and implementing complex Fourier multiplication directly with `einsum`, yield a notable boost in accuracy, speed and parameter counts. We refer to this refined baseline as FNO$^+$, and it surpasses many neural operator variants introduced after the original FNO. The appendix provides a detailed breakdown of the specific changes made to FNO. Note that, D-SENO and FNO$^+$ have been trained using the same hyper parameters as shown in Table 3, unless stated otherwise.

Table 2: Cross-representation test errors (lower is better). '/' indicates a value not reported.

| MODEL | STRUCTURED MESH | | REGULAR GRID | |
|---|---|---|---|---|
| | AIRFOIL | PIPE | DARCY | NAVIER–STOKES |
| FNO Li et al. (2020) | / | / | 0.0108 | 0.1556 |
| WMT Gupta et al. (2021) | 0.0075 | 0.0077 | 0.0082 | 0.1541 |
| F-FNO Tran et al. (2021) | 0.0078 | 0.0070 | 0.0077 | 0.2322 |
| U-NO Rahman et al. (2022) | 0.0078 | 0.0100 | 0.0113 | 0.1713 |
| U-FNO Wen et al. (2022) | 0.0269 | 0.0056 | 0.0183 | 0.2231 |
| geo-FNO Li et al. (2023b) | 0.0138 | 0.0067 | 0.0108 | 0.1556 |
| LSM Wu et al. (2023) | 0.0059 | 0.0050 | 0.0065 | 0.1535 |
| Galerkin Cao (2021) | 0.0118 | 0.0098 | 0.0084 | 0.1401 |
| HT-Net Liu et al. (2022) | 0.0065 | 0.0059 | 0.0079 | 0.1847 |
| OFormer Li et al. (2022) | 0.0183 | 0.0168 | 0.0124 | 0.1705 |
| GNOT Hao et al. (2023) | 0.0076 | 0.0047 | 0.0105 | 0.1380 |
| FactFormer Li et al. (2023a) | 0.0071 | 0.0060 | 0.0109 | 0.1214 |
| ONO Xiao et al. (2023) | 0.0061 | 0.0052 | 0.0076 | 0.1195 |
| FNO$^+$ (Time s) | 0.0057 (1.67) | 0.0072 (5.41) | 0.0070 (1.10) | 0.1054 (5.75) |
| Transolver (Time s) | 0.0053 (28.03) | 0.0033 (41.00) | 0.0057 (18.47) | **0.0900** (**245.10**) |
| **Ours** (Time s) | **0.0052** (**1.99**) | **0.0030** (**3.30**) | **0.0048** (**0.93**) | 0.1391 (7.04) |

All D-SENO experiments are performed on a single NVIDIA A100 80 GB PCIe GPU, and performance numbers are averaged over three independent random seeds. We report the relative $L_2$ error of predicted fields on the held out test sets as the primary evaluation metric. For additional metrics and ablation results along with the hyperparameter details for every benchmark, we refer the reader to supplementary materials.

## 6 Results and Discussion

Table 2 summarizes the results obtained by D-SENO on the PDE benchmarks. The model consistently outperforms the prior state-of-the-art Transolver Wu et al. (2024) on three out of four datasets, showing markedly reduced training time ("Time s" in Table 2 represents the training time per epoch) along with superior predictive accuracy, highlighting the high accuracy and lightweight design of D-SENO. Conducting similar experiments with FNO$^+$, we find that it consistently achieves significantly higher accuracy than the original FNO in all four datasets. Moreover, it outperforms many neural operators developed after FNO on the same benchmarks. For details on how its performance varies with the number of Fourier modes and the projection width, we refer the reader to the supplementary material.

Figure 3 compares the Mach number reconstructions around a deformed NACA–0012 airfoil in latent space. Both D-SENO and FNO$^+$ capture the overall shock and expansion pattern, but FNO$^+$ shows slightly larger errors, softening those steep Mach transitions. D-SENO, in contrast, significantly resolves these discontinuities, closely matching the reference peak and trough values along the airfoil surface.

Meanwhile, Figure 4 presents the centerline velocity profiles for Poiseuille pipe flow. Although both operators reproduce the characteristic velocity profile, FNO$^+$ exhibits higher errors continuously along the edges of the pipe, indicating a persistent under prediction of the steep velocity gradient. D-SENO, by contrast, confines its largest discrepancies to the downstream end of the pipe and at significantly lower magnitudes than FNO$^+$, thereby preserving accurate gradient structure throughout the channel.

Figure 5 shows the results for Darcy flow, it can be observed that FNO$^+$ tends to systematically over predict the pressure in regions of high permeability, resulting in larger relative errors. In contrast, D-SENO exhibits minimal underestimation overall. This behavior further highlights D-SENO's superior ability to resolve sharp transitions and fine-scale heterogeneity in complex porous media flows.

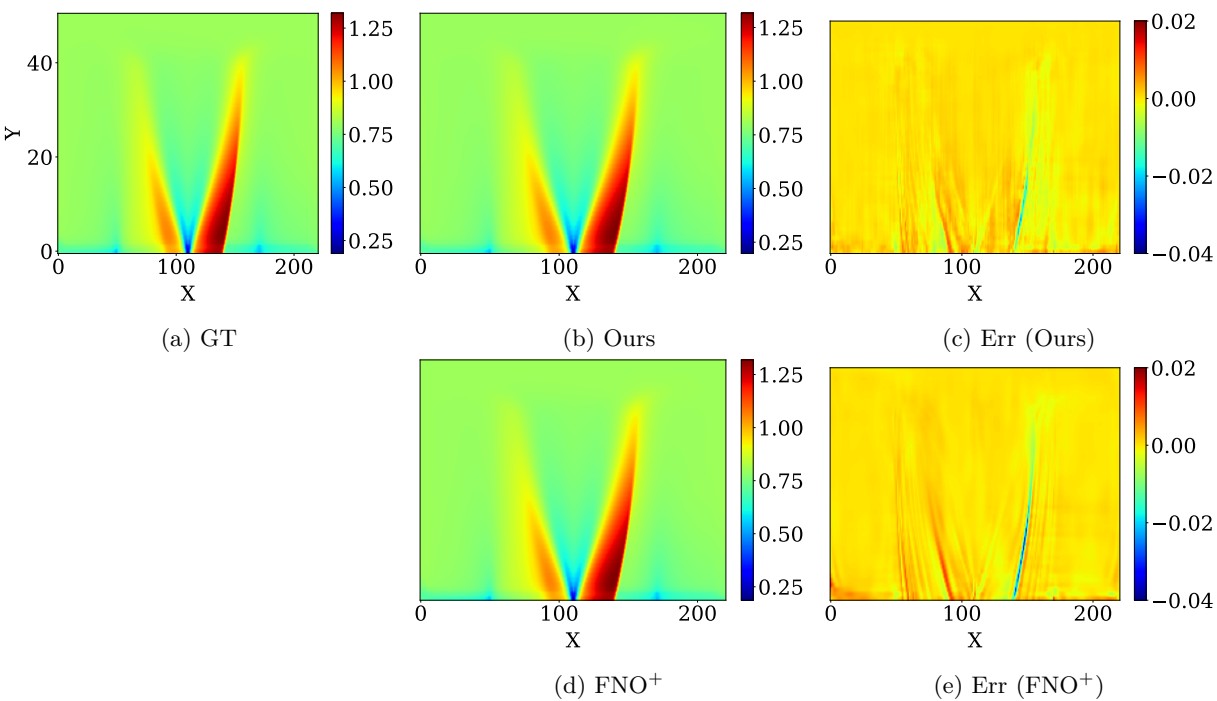

Figure 3: Sample from the Airfoil dataset.

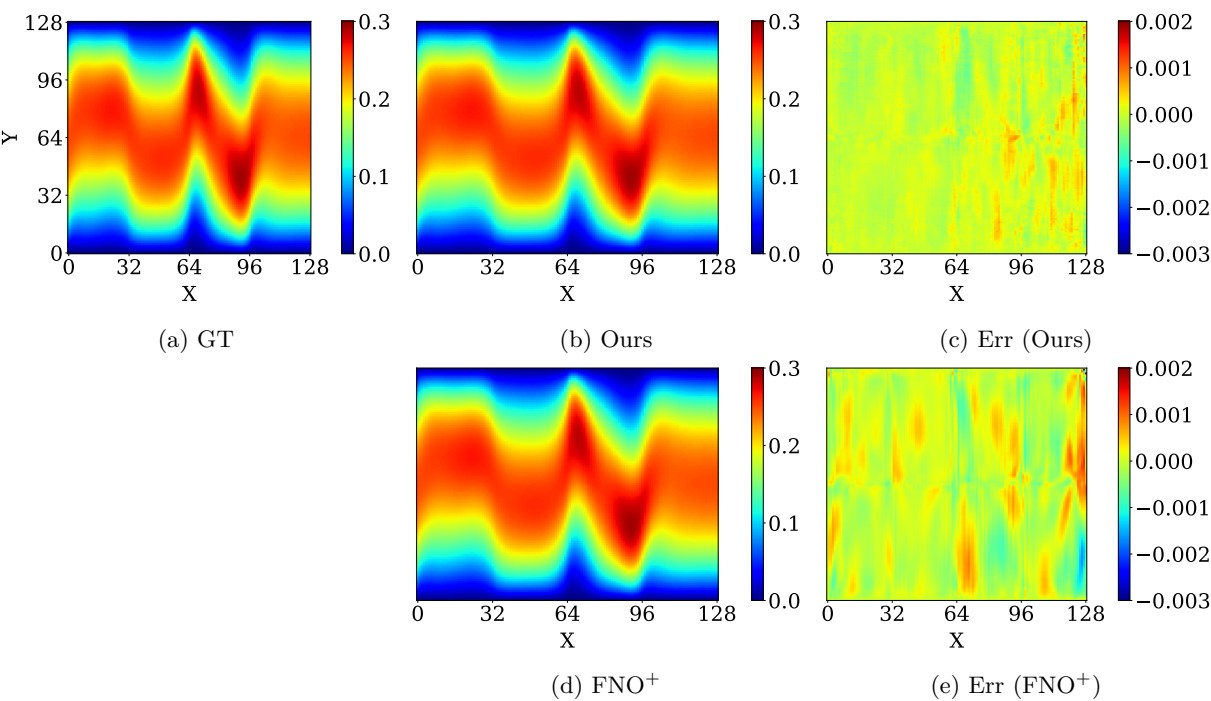

Figure 4: Sample from the Pipe dataset.

In the Navier–Stokes torus dataset shown in Figure 6, 7, FNO$^+$ outperforms D-SENO. This may be due to periodic boundary conditions on the unit torus, that align naturally with FNO's Fourier-based representation, enabling it to capture spatiotemporal flow patterns more effectively. Moreover, the 2D+T forecasting task leverages FNO$^+$'s ability to perform multi-step predictions in the frequency domain. In contrast, D-SENO

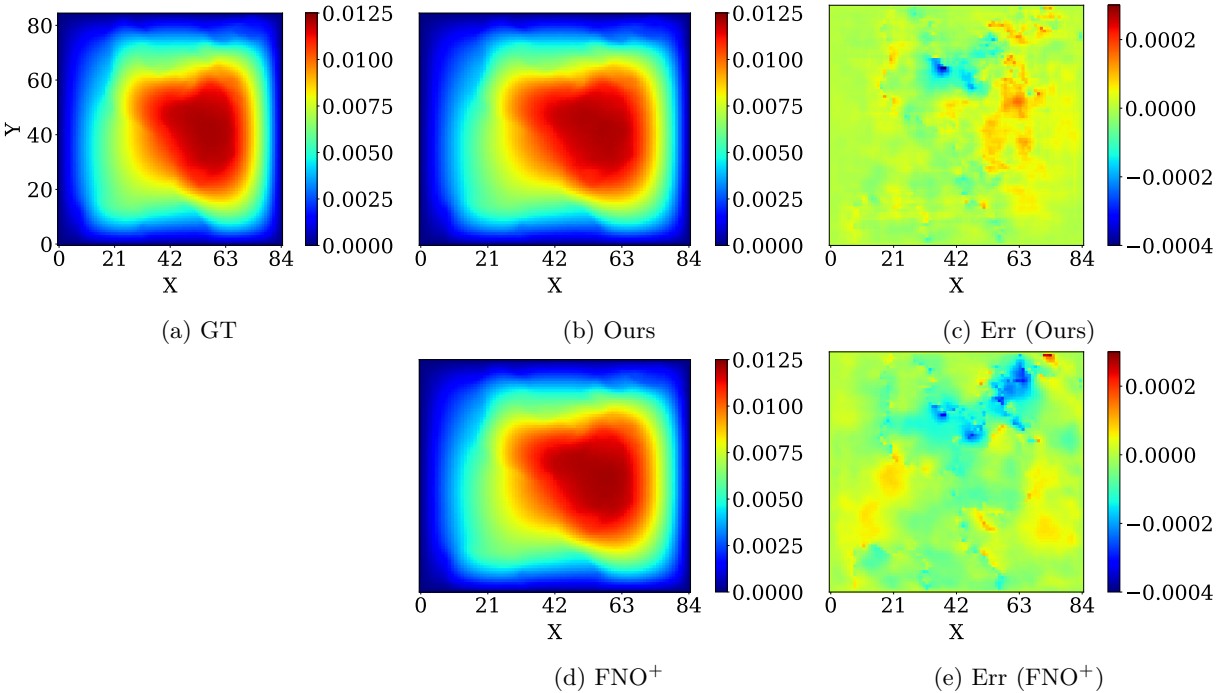

(a) GT                (b) Ours                (c) Err (Ours)

(d) FNO$^+$                (e) Err (FNO$^+$)

Figure 5: Sample from the Darcy dataset.

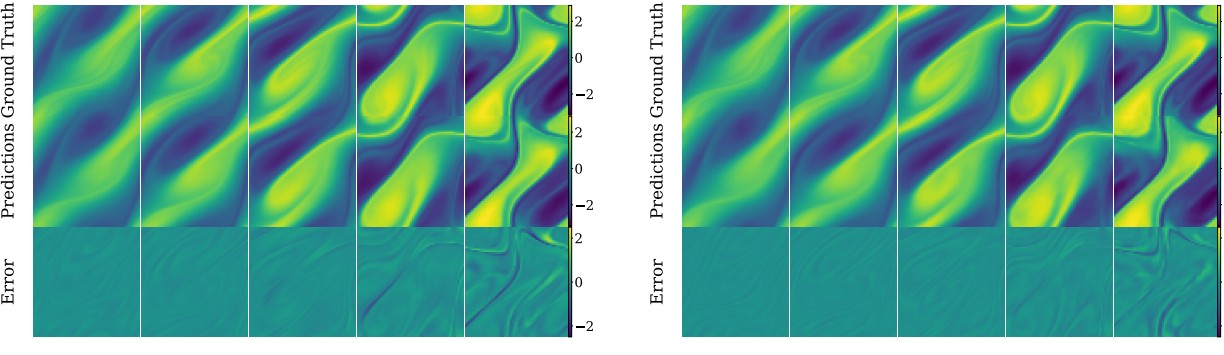

Figure 6: Predicted D-SENO results for Navier–Stokes fields at times $t = 12, 14, 16, 18$, and 20 (left to right).

Figure 7: Predicted FNO$^+$ results for Navier–Stokes fields at times $t = 12, 14, 16, 18$, and 20 (left to right).

primarily performs better for static spatial deformations and does not explicitly exploit temporal periodicity, which likely limits its accuracy in this dataset. Future extensions, such as integrating periodic temporal kernels or expanding D-SENO's spectral receptive field may help bridge this gap.

These results collectively demonstrate that D-SENO captures discontinuities better than FNO$^+$ not only in shock-dominated airfoil flows and shear-layer pipe flows, but also in Darcy porous-media flows. By using multiple rates of dilated convolutions to expand its receptive field, aggregating multiscale context, and using squeeze-and-excitation blocks to adaptively re-weight channel responses, D-SENO sharpens and emphasizes steep gradients, preserving them with lower local error and resolving fine-scale heterogeneity in the solution.

# 7 Conclusion

This paper introduces the D-SENO , a fully convolution surrogate that combines receptive-field expanding dilated kernels with lightweight squeeze and excitation channel attention. By retaining strict locality, that is, no Fourier transforms, self-attention, or positional encoding, D-SENO reduces the time complexity for training the model while capturing domain-scale interactions that are essential in fluid and porous-media flows. Extensive experiments on canonical PDE benchmarks show that D-SENO attains low relative error while delivering speed-ups of multiple orders of magnitude over transformer-based solvers and neural operators. Beyond its empirical performance, D-SENO also offers practical benefits as it relies exclusively on standard convolution operations, enabling efficient execution on existing, highly optimized hardware.

**Limitations.** While D-SENO achieves strong accuracy and efficiency trade-offs on a range of benchmarks, several aspects merit further development. First, on the Navier-Stokes benchmark, transformer-based solvers and neural operators often attain higher accuracy, suggesting that more explicit global context aggregation may be beneficial for highly multiscale dynamics. Second, our evaluation focuses on structured-grid datasets, extending the framework to unstructured meshes and more complex geometries is an important next step to broaden applicability. Third, the present study is primarily empirical, and a complementary theoretical analysis of approximation, stability, and generalization would strengthen understanding. Finally, dilation schedules are currently chosen per dataset, learning or adapting dilation patterns automatically could improve robustness, with potential compute and accuracy trade-offs.

**Future work.** Promising avenues for future work include extending D-SENO to unstructured meshes via masked or graph-based dilation, integrating physics-informed priors or constraints to improve generalization and combining D-SENO with attention modules to capture long-range dependencies even better. We envision D-SENO as a simple, computationally efficient baseline and a flexible building block for the next generation of neural solvers in science and engineering.

# 8 Acknowledgment

This work was supported by the NYUAD Center for Interacting Urban Networks (CITIES), funded by Tamkeen under the NYUAD Research Institute Award CG001. The views expressed in this article are those of the authors and do not reflect the opinions of CITIES or their funding agencies.

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

Table 3: Training hyperparameters for each benchmark.

| Benchmark | $n_{\text{train}}$ | $n_{\text{test}}$ | Width | Loss | Epochs | Batch | LR | Step | Decay | Weight Decay |
|---|---|---|---|---|---|---|---|---|---|---|
| Airfoil | 1000 | 200 | 64 | Rel. $L_2$ | 500 | 20 | $1 \times 10^{-3}$ | 100 | 0.5 | $1 \times 10^{-4}$ |
| Pipe | 1000 | 200 | 96 | Rel. $L_2$ | 500 | 20 | $1 \times 10^{-3}$ | 100 | 0.5 | $1 \times 10^{-4}$ |
| Darcy | 1000 | 200 | 48 | Rel. $L_2$ | 500 | 20 | $1 \times 10^{-3}$ | 100 | 0.5 | $1 \times 10^{-4}$ |
| Navier–Stokes | 1000 | 200 | 64 | Rel. $L_2$ | 500 | 20 | $1 \times 10^{-3}$ | 100 | 0.5 | $1 \times 10^{-4}$ |

Table 4: Model parameterization for each benchmark.

| Dataset | DC Block Conv 1 Kernel | Bias | DC Block Conv 2 Kernel | Bias | SE Reduction | SE Block Conv 1 Kernel | Bias | SE Block Conv 2 Kernel | Bias |
|---|---|---|---|---|---|---|---|---|---|
| Airfoil | $3 \times 3$ | Yes | $5 \times 5$ | No | 1 | $1 \times 1$ | Yes | $1 \times 1$ | Yes |
| Pipe | $3 \times 3$ | Yes | $3 \times 3$ | Yes | 1 | $1 \times 1$ | Yes | $1 \times 1$ | Yes |
| Darcy | $3 \times 3$ | Yes | $5 \times 5$ | No | 1 | $1 \times 1$ | Yes | $1 \times 1$ | Yes |
| Navier–Stokes | $3 \times 3$ | Yes | $3 \times 3$ | Yes | 1 | $1 \times 1$ | Yes | $1 \times 1$ | Yes |

Haixu Wu, Huakun Luo, Haowen Wang, Jianmin Wang, and Mingsheng Long. Transolver: A fast transformer solver for pdes on general geometries. *arXiv preprint arXiv:2402.02366*, 2024.

Zipeng Xiao, Zhongkai Hao, Bokai Lin, Zhijie Deng, and Hang Su. Improved operator learning by orthogonal attention. *arXiv preprint arXiv:2310.12487*, 2023.

Bo Xu, Xinliang Liu, and Lei Zhang. Dilated convolution neural operator for multiscale partial differential equations. *Journal of Computational and Applied Mathematics*, 461:116408, 2025.

Fisher Yu and Vladlen Koltun. Multi-scale context aggregation by dilated convolutions. *arXiv preprint arXiv:1511.07122*, 2015.

# A  Appendix

## A.1  Ablation Study

Our ablation methodology addresses five architectural and data-related parameters: (i) depth (number of DS blocks), (ii) squeeze and excitation block, (iii) dilation configuration, (iv) projection width (latent channel dimensionality), and (v) dataset resolution (for Darcy dataset only). We apply this study to our model D-SENO (Dilated Squeeze-and-Excitation Neural Operator), FNO, the FNO$^+$ variant introduced in the main paper, and Transolver, and discuss each parameter in detail in the following subsections. Table 3 and Table 4 details the training configuration (optimizer type, learning rate and schedule, batch size, number of epochs, etc.) and the model configurations used for each of the datasets in the following experiments. Note that D-SENO, FNO$^+$ and FNO use the same parameters, while for Transolver the parameters given in Wu et al. (2024) were used unless otherwise specified. The datasets used in this study are publicly available: airfoil potential flow and Poiseuille pipe flow datasets are available at `https://github.com/neuraloperator/Geo-FNO`, and the Darcy flow and Navier–Stokes datasets are available at `https://drive.google.com/drive/folders/1UnbQh2WWc6knEHbLn-ZaXrKUZhp7pjt-`.

### A.1.1  Depth Sensitivity

In this subsection, we evaluate how model depth, defined by the number of DS blocks, affects the predictive accuracy. For each of the three steady benchmark datasets (Airfoil, Pipe and Darcy dataset), we systematically increase the number of DS blocks to quantify accuracy gains as model depth grows, while keeping all other architectural and training settings fixed. At each depth, we record the complete configuration and report the time per epoch (in seconds), the number of trainable parameters (in millions), and the relative $L_2$ error; all error values are averaged over multiple random seeds to mitigate variance. In addition to our

Table 5: Model performance summary on the Airfoil dataset.

| Model | # DS Blocks | Time per Epoch (s) | # Parameters (M) | Relative $L_2$ Error |
|---|---|---|---|---|
| Model Airfoil-A | 1 | 0.43 | 0.156 | 0.0131 |
| Model Airfoil-B | 2 | 0.67 | 0.304 | 0.0072 |
| Model Airfoil-C | 3 | 0.93 | 0.451 | 0.0067 |
| Model Airfoil-D | 4 | 1.16 | 0.599 | 0.0057 |
| Model Airfoil-E | 5 | 1.43 | 0.747 | 0.0055 |
| Model Airfoil-F | 6 | 1.68 | 0.894 | 0.0055 |
| **Model Airfoil-G** | 7 | 1.99 | 1.042 | **0.0052** |
| Model Airfoil-G w/o SE | 7 | 1.70 | 0.984 | 0.0056 |
| Model Airfoil-G w/o SE (PM) | 7 | 2.05 | 1.042 | 0.0056 |
| Model Airfoil-G-alt | 7 | 2.04 | 1.042 | 0.0054 |
| Transolver | – | 28.03 | 2.811 | 0.0053 |
| FNO$^+$ ($m = 8$) | – | 1.39 | 2.123 | 0.0058 |
| FNO$^+$ ($m = 16$) | – | 1.51 | 8.414 | 0.0059 |
| FNO$^+$ ($m = 24$) | – | 1.67 | 18.899 | 0.0057 |
| FNO Original ($m = 24$) | – | 2.47 | 37.774 | 0.0060 |

Table 6: Detailed dilation values for Airfoil dataset showing $d_x$ and $d_y$ in each model.

| Model | $d_x$ | $d_y$ |
|---|---|---|
| Model Airfoil-A | [16] | [6] |
| Model Airfoil-B | [16,54] | [4,10] |
| Model Airfoil-C | [16,48,2] | [1,6,4] |
| Model Airfoil-D | [16,56,30,2] | [1,2,10,4] |
| Model Airfoil-E | [16,56,36,24,1] | [1,2,10,6,1] |
| Model Airfoil-F | [16,56,42,36,24,1] | [1,2,8,12,6,1] |
| Model Airfoil-G | [16,56,42,36,32,24,1] | [1,2,8,12,6,2,1] |
| Model Airfoil-G w/o SE | [16,56,42,36,32,24,1] | [1,2,8,12,6,2,1] |
| Model Airfoil-G w/o SE (PM) | [16,56,42,36,32,24,1] | [1,2,8,12,6,2,1] |
| Model Airfoil-G-alt | [20,52,46,38,30,20,2] | [2,4,8,10,8,4,2] |

model, we also report the above metrics for Transolver, FNO and FNO$^+$. We test FNO$^+$ by varying the number of Fourier modes, then test FNO using the modes count that gave the best results for FNO$^+$.

The results for airfoil, pipe, and Darcy dataset are summarized in Tables 5, 7 and 9. Note that in the tables, each model variant is denoted by appending successive letters (A, B, C, . . . ) to its name, corresponding to increasing numbers of DS blocks. The results show that increasing the number of DS blocks lowers the relative $L_2$ error, this is because the additional dilation rates expand the receptive field and yield significant accuracy gains, but this comes with increased computational cost, as can be seen by the increased number of parameters and higher training time per epoch. It can also be seen that FNO$^+$ significantly outperforms the regular FNO in all cases, showing that its architectural enhancements, increase its expressive capacity to capture the underlying dynamics. Similar experiments on the time-varying Navier-Stokes dataset, shown in Table 11, were carried out with only a single dilation rate, since D-SENO did not outperform FNO$^+$. The details of the exact dilation rates used for the above experiments for each of the four dataset, are given in Table 6, 8, 10 and 12 where $d_x$ gives the dilation rates along x dimension and $d_y$ gives the dilation rates along the y dimension.

### A.1.2 Squeeze and Excitation impact

In this subsection, we assess the impact of the Squeeze-and-Excitation (SE) block by comparing the full D-SENO to a variant with the SE block removed from each DS block. The ablated model is identified in the results table by the suffix "w/o SE" appended to its name. From Tables 5, 7, 9 and 11, it can be observed that omitting the SE block leads to a substantial degradation in D-SENO's accuracy, reflected in a significant increase in the relative $L_2$ error, which highlights the importance of the SE module for effective feature representation learning.

Table 7: Model performance summary on the Pipe dataset.

| Model | # DS Blocks | Time / Epoch (s) | Parameters (M) | Relative $L_2$ Error |
|---|---|---|---|---|
| Model Pipe-A | 1 | 0.77 | 0.197 | 0.0107 |
| Model Pipe-B | 2 | 1.29 | 0.382 | 0.0073 |
| Model Pipe-C | 3 | 1.79 | 0.567 | 0.0046 |
| Model Pipe-D | 4 | 2.31 | 0.752 | 0.0040 |
| Model Pipe-E | 5 | 2.80 | 0.936 | 0.0035 |
| Model Pipe-F | 6 | 3.47 | 1.121 | 0.0031 |
| **Model Pipe-G** | 7 | 4.28 | 1.306 | **0.0030** |
| Model Pipe-G w/o SE | 7 | 3.30 | 1.176 | 0.0030 |
| Model Pipe-G w/o SE (PM) | 7 | 4.01 | 1.306 | 0.0037 |
| Model Pipe-G-alt | 7 | 4.05 | 1.306 | 0.0030 |
| Transolver | – | 40.01 | 2.811 | 0.0033 |
| FNO$^+$ $(m = 8)$ | – | 5.41 | 4.769 | 0.0074 |
| FNO$^+$ $(m = 16)$ | – | 5.79 | 18.925 | 0.0075 |
| FNO$^+$ $(m = 32)$ | – | 6.72 | 75.548 | 0.0072 |
| FNO Original $(m = 32)$ | – | 4.44 | 9.488 | 0.0086 |

Table 8: Detailed dilation values for Pipe dataset showing $d_x$ and $d_y$ in each model.

| Model | $d_x$ | $d_y$ |
|---|---|---|
| Model Pipe-A | [9] | [9] |
| Model Pipe-B | [23,1] | [23,1] |
| Model Pipe-C | [23,11,1] | [23,11,1] |
| Model Pipe-D | [23,15,7,1] | [23,15,7,1] |
| Model Pipe-E | [23,15,9,3,1] | [23,15,9,3,1] |
| Model Pipe-F | [25,19,11,7,3,1] | [25,19,11,7,3,1] |
| Model Pipe-G | [23,17,13,9,7,3,1] | [23,17,13,9,7,3,1] |
| Model Pipe-G w/o SE | [23,17,13,9,7,3,1] | [23,17,13,9,7,3,1] |
| Model Pipe-G w/o SE (PM) | [23,17,13,9,7,3,1] | [23,17,13,9,7,3,1] |
| Model Pipe-G-alt | [25,19,11,9,5,3,1] | [25,19,11,9,5,3,1] |

Additionally, we conduct a parameter-matched ablation to separate the effect of the SE mechanism from that of model capacity. In this variant, the SE block is still removed, but the parameter count is restored to match the SE-based D-SENO by appending two $1 \times 1$ point-wise convolutions after the dilated spatial convolutions in each DC block. This adds learnable channel-mixing capacity while preserving the original receptive field and residual structure, yet does not perform explicit SE-style channel recalibration. We denote this model as "w/o SE (PM)" in the results, so that performance differences relative to the full model primarily reflect the functional role of SE block rather than reduced parameters.

From the results shown in Tables 5, 7, 9 and 11, we see that the SE-based model consistently achieves better performance than the parameter-matched baseline on the airfoil, pipe, and Navier–Stokes datasets, while the baseline obtained by simply increasing the number of parameters performs better only on the Darcy flow task. This pattern indicates that the performance gains largely stem from the SE architecture rather than mere model size, and that overall our SE-based approach provides the stronger model across datasets.

### A.1.3 Dilation rate impact

In this study, the dilation rates are dataset-specific hyperparameters rather than learned quantities, selected via a small number of manual trials using a simple, well-spaced pattern that covers both local and global spatial scales. This is a heuristic choice, not an exhaustive grid search or an adaptive method. To isolate the influence of receptive-field enlargement strategy, we replace the adopted dilation rates in the SE-based model with an alternative rates for each dataset, keeping all other hyperparameters constant, this reveals the sensitivity of the model to the chosen pattern of dilation. Models employing these alternative rates are identified by the suffix "-alt" appended to their names in the results. The exact dilation rates used are given

Table 9: Model performance summary on the Darcy dataset.

| Model | # DS Blocks | Time / Epoch (s) | Parameters (M) | Relative $L_2$ Error |
|---|---|---|---|---|
| Model Darcy-A | 1 | 0.35 | 0.089 | 0.0282 |
| Model Darcy-B | 2 | 0.46 | 0.173 | 0.0247 |
| Model Darcy-C | 3 | 0.55 | 0.256 | 0.0084 |
| Model Darcy-D | 4 | 0.69 | 0.338 | 0.0064 |
| Model Darcy-E | 5 | 0.80 | 0.422 | 0.0051 |
| Model Darcy-F | 6 | 0.93 | 0.505 | 0.0048 |
| Model Darcy-F w/o SE | 6 | 0.80 | 0.476 | 0.0053 |
| **Model Darcy-F w/o SE (PM)** | 6 | 0.99 | 0.505 | **0.0046** |
| Model Darcy-F-alt | 6 | 0.92 | 0.505 | 0.0050 |
| Transolver | – | 18.48 | 2.811 | 0.0057 |
| FNO$^+$ ($m = 8$) | – | 0.78 | 1.196 | 0.0086 |
| FNO$^+$ ($m = 16$) | – | 0.90 | 4.735 | 0.0073 |
| FNO$^+$ ($m = 32$) | – | 1.10 | 18.890 | 0.0070 |
| FNO$^+$ ($m = 42$) | – | 1.30 | 32.531 | 0.0071 |
| FNO Original ($m = 32$) | – | 1.92 | 37.765 | 0.0089 |

Table 10: Detailed dilation values for Darcy dataset showing $d_x$ and $d_y$ in each model.

| Model | $d_x$ | $d_y$ |
|---|---|---|
| Model Darcy-A | [7] | [7] |
| Model Darcy-B | [1,19] | [1,19] |
| Model Darcy-C | [1,11,19] | [1,11,19] |
| Model Darcy-D | [1,7,13,19] | [1,7,13,19] |
| Model Darcy-E | [1,5,9,13,19] | [1,5,9,13,19] |
| Model Darcy-F | [1,3,5,9,13,19] | [1,3,5,9,13,19] |
| Model Darcy-F w/o SE | [1,3,5,9,13,19] | [1,3,5,9,13,19] |
| Model Darcy-F w/o SE (PM) | [1,3,5,9,13,19] | [1,3,5,9,13,19] |
| Model Darcy-F-alt | [1,3,7,11,15,21] | [1,3,7,11,15,21] |

in Table 6, 8, 10 and 12. It can be observed that, across the datasets, using alternative dilation rates results in only negligible changes in accuracy, showing the robustness of the model.

### A.1.4 Projection width

In this case, we vary the projection width of the inputs to examine the trade-off between representational capacity and performance of the best performing model from Tables 5, 7 and 9. The results are shown in Figure 8. It can be observed that the relative $L_2$ loss decreases with projection width until it reaches a dataset specific minimum. Beyond this width, further increases produce negligible improvement. Also, as projection width increases, both time per epoch and parameter count grow, reflecting the additional computational cost.

### A.1.5 Resolution robustness

To probe resolution robustness, we used heterogeneous Darcy flow dataset. The dataset was uniformly down sampled from the original $421 \times 421$ grid to several spatial resolutions. Each of these resolutions were used to test the three models : Transolver, FNO$^+$ and D-SENO as shown in Table 13. The results indicate that D-SENO outperforms both Transolver and FNO$^+$ across all resolutions. The exact rates of dilation used for D-SENO are given in Table 15. Complementing these findings, Table 14 reports a systematic analysis of how varying the number of retained Fourier modes influences performance at each resolution for FNO$^+$. Note that the hyper parameters and error used here for Transolver are the same as mentioned in Wu et al. (2024) for Darcy dataset.

Table 11: Model performance summary on the Navier–Stokes dataset.

| Model | # DS Blocks | Time / Epoch (s) | Parameters (M) | Relative $L_2$ Error |
|---|---|---|---|---|
| Model NS-A | 8 | 7.04 | 0.666 | 0.1391 |
| Model NS-A w/o SE | 8 | 5.94 | 0.600 | 0.1432 |
| Model NS-A w/o SE (PM) | 8 | 7.53 | 0.666 | 0.1433 |
| Model NS-A-alt | 8 | 7.67 | 0.666 | 0.1431 |
| Transolver | – | 245.10 | 11.232 | **0.0900** |
| FNO$^+$ $(m=8)$ | – | 5.75 | 2.123 | 0.1054 |
| FNO$^+$ $(m=16)$ | – | 5.81 | 8.414 | 0.1122 |
| FNO$^+$ $(m=32)$ | – | 6.93 | 33.580 | 0.1153 |
| FNO Original $(m=8)$ | – | 13.53 | 4.220 | 0.1436 |

Table 12: Dilation values ($d_x$ and $d_y$) for models corresponding to performance results in Table 11.

| Model | $d_x$ | $d_y$ |
|---|---|---|
| Model NS-A | [15,25,17,13,7,5,3,1] | [15,25,17,13,7,5,3,1] |
| Model NS-A w/o SE | [15,25,17,13,7,5,3,1] | [15,25,17,13,7,5,3,1] |
| Model NS-A w/o SE (PM) | [15,25,17,13,7,5,3,1] | [15,25,17,13,7,5,3,1] |
| Model NS-A-alt | [21,27,19,11,9,7,3,1] | [21,27,19,11,9,7,3,1] |

## A.2 FNO$^+$ Architecture

We introduce FNO$^+$ as a minimally modified variant of the original FNO, designed to serve as a strong and improved baseline. In its original formulation, the FNO architecture consists of a lifting layer that maps the input fields to a higher-dimensional channel space, followed by a stack of Fourier layers and a projection layer that maps back to the target field. Each Fourier layer performs a spectral convolution with the retained Fourier modes, combined with a point-wise linear transformation in physical space. The default implementation uses ReLU activations and includes batch-normalization layers within each Fourier block, while realizing complex Fourier multiplications via separate real and imaginary linear projections. Architecturally, FNO$^+$ has the following changes compared to the standard FNO: (i) we replace the ReLU nonlinearity with GELU in all FNO layers, (ii) we remove all batch-normalization layers, and (iii) we implement complex-valued Fourier multiplication directly using a single einsum-based operation. Beyond these changes, the layer types, block structure, and connectivity patterns are kept identical to the original FNO design. On top of this fixed architecture, we consider purely experimental variations, reported in the appendix, such as runs with different numbers of retained Fourier modes and different widths in the lifting layer to probe the effect of spectral resolution and model capacity.

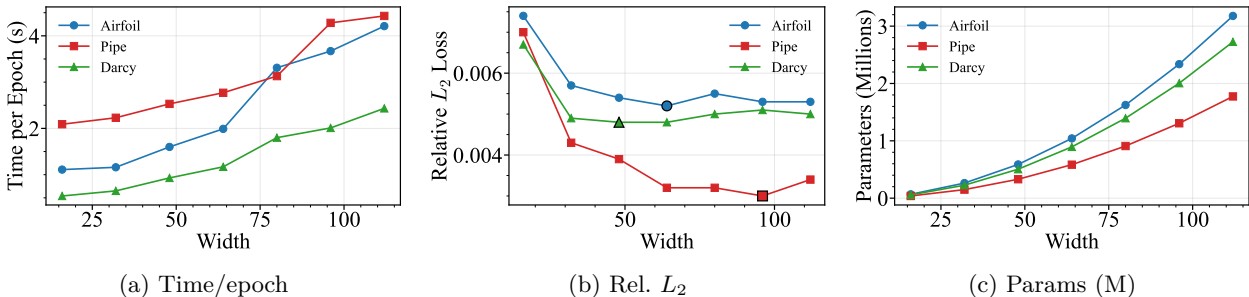

Figure 8: Performance metrics vs. projection width.

Table 13: Performance metrics (time per epoch, parameter count, and relative $L_2$ error) for different models at various input resolutions on the Darcy dataset.

| Model | $32 \times 32$ | | | $64 \times 64$ | | | $128 \times 128$ | | | $256 \times 256$ | | |
|---|---|---|---|---|---|---|---|---|---|---|---|---|
| | Time (s) | Params (M) | Rel. $L_2$ | Time (s) | Params (M) | Rel. $L_2$ | Time (s) | Params (M) | Rel. $L_2$ | Time (s) | Params (M) | Rel. $L_2$ |
| Transolver | 12.21 | 2.827 | 0.0135 | 12.56 | 2.827 | 0.0069 | 38.95 | 2.827 | 0.0052 | 151.73 | 2.827 | 0.0058 |
| FNO$^+$ | 0.66 | 4.734 | 0.0159 | 0.74 | 18.890 | 0.0083 | 2.33 | 75.513 | 0.0065 | 9.52 | 302.006 | 0.0063 |
| D-SENO | 0.35 | 0.256 | **0.0131** | 0.70 | 0.588 | **0.0063** | 1.73 | 0.505 | **0.0042** | 7.96 | 0.588 | **0.0036** |

Table 14: 2D Darcy relative $L_2$ loss and per-epoch time vs. number of modes across resolutions for FNO$^+$.

| MODES | $32 \times 32$ | | $64 \times 64$ | | $128 \times 128$ | | $256 \times 256$ | | PARAMS (M) |
|---|---|---|---|---|---|---|---|---|---|
| | Rel $L_2$ | TIME (s) | Rel $L_2$ | TIME (s) | Rel $L_2$ | TIME (s) | Rel $L_2$ | TIME (s) | |
| 8 | 0.0157 | 0.65 | 0.0092 | 0.66 | 0.0081 | 1.19 | 0.0080 | 5.32 | 1.195 |
| 16 | **0.0159** | 0.66 | 0.0084 | 0.67 | 0.0070 | 1.23 | 0.0068 | 4.96 | 4.734 |
| 32 | – | – | **0.0083** | 0.74 | 0.0067 | 1.44 | 0.0065 | 5.85 | 18.890 |
| 64 | – | – | – | – | **0.0065** | 2.33 | 0.0064 | 5.68 | 75.513 |
| 128 | – | – | – | – | – | – | **0.0063** | 9.52 | 302.006 |

Table 15: Dilation values ($d_x$ and $d_y$) for models corresponding to performance results in Table 13.

| Resolution | $d_x$ | $d_y$ |
|---|---|---|
| $32 \times 32$ | [1,2,6] | [1,2,6] |
| $64 \times 64$ | [1,3,5,7,9,13,15] | [1,3,5,7,9,13,15] |
| $128 \times 128$ | [1,5,9,15,21,27] | [1,5,9,15,21,27] |
| $256 \times 256$ | [1,5,7,15,23,39,61] | [1,5,7,15,23,39,61] |

