# OpenReview forum: "Efficient Dilated Squeeze and Excitation Neural Operator for Differential Equations"
_TMLR — Accepted by TMLR_

### Review · Reviewer_HWMT · 2025-10-28

**Summary Of Contributions:**

Current PDE surrogates lead to expensive training and slow inference due to the large number of parameters in the models. Applications of PDE surrogates require models that are fast and accurate. The paper presents a lightweight operator learning framework for solving a range of PDEs while achieving up to 20x speedup over standard transformer-based models and neural operators. It also matches or improves upon the accuracy of PDE solving.

The paper identifies limitations of prior work that are fixed by the proposal. Fourier transform based operators struggle with localized features due to lack of spatial localization in spectral kernels. Convolutional neural operators do not adaptively enlarge receptive fields and dilated convolutional operators are limited to periodic inputs due to the use of the FFT.

The method used is dilated convolution (DC) blocks with squeeze-and-excitation (SE) modules that jointly capture e wide receptive fields and dynamics in addition to channel-wise attention. Dilation rates allow the receptive field to focus on critical regions to model long range dependencies.
The dilations allow the receptive field to grow without downsampling or increasing the number of parameters.
The SE block adjusts the importance of each feature channel. It uses global average pooling to compute channel-wise statistics which are transformed into scaling weights and applied with element-wise multiplication.
The blocks are combined into a DS layer which begins with a dilated conv block followed by a SE block. The full model follows a lifting, processing and project setup and if fully convolutional.

Experiments are performed on four benchmarks datasets (Airfoil, Pipe, Darcy and Navier Stokes) also used by previous operator approaches, spanning structured and regular grids. From Table 2 test errors are on-par or better than the transformer baseline except for the Navier-stokes dataset. For the other datasets performance appears to be on-par with the transformer model with faster inference time. The performance in inference time appears to be on-par with FNO and is worse in some cases. Accuracy is better than FNO also except for the Navier-Stokes dataset where it is worse on both counts (accuracy and time).

In the discussion (second last paragraph page 8) the paper states “... higher accuracy than the original FNO on all three datasets”. I wonder why this does not count the Navier-Stokes dataset in table 2?

**Audience:**

Yes

**Audience Explanation:**

The model presents a simple PDE surrogate model which combines architectural features (Dilated convolutions, squeeze-and-excitation block) to develop a lightweight operator model. This should be of interest to researchers in efficient PDE surrogate modeling.

**Broader Impact Concerns:**

No concerns.

**Claims And Evidence:**

Yes

**Claims Explanation:**

The main claim of the paper is a method that is lightweight, fast and accurate for PDE surrogate modeling tasks. Speed and accuracy are shown by experiments on four datasets where the method is shown to be significantly faster than the transformer models with similar accuracy on three out of the four datasets. The performance on the fourth dataset is somewhat worse in accuracy but better in time.

The model is lightweight by being fully convolutional and has significantly fewer parameters, shown in Table 13.

**Requested Changes:**

Section 4.2 says “these layers reduce and restore the number of channels…”. Explanation as to how and where this is done should be included at this point.

Could you add an explanation of dilation rates and adaptive receptive fields in the main paper itself.? In what sense are these adaptive if they have to be chosen per dataset?

Can you explain why you don’t consider all four datasets when summarizing the performance comparison with FNO in section 6 paragraph 1?

Typos

Section 2. DNCO should not be bold.
In section 4.1 G(a) = u should be written as u = G(a)
How is the inference time in Table 2 computed? Is this per example? per batch?

---

> ### Author Response · Authors · 2025-12-14
> **Response to Reviewer HWMT**
>
> We thank the reviewer for their constructive feedback.
>
> **Response regarding typo (Discussion, second last paragraph, page 8):** Indeed, $\mathrm{FNO}^{+}$ outperformed original FNO in all four datasets including Navier-Stokes, we have corrected this typo in the revised manuscript.
>
> **Response regarding section 4.2, “these layers reduce and restore the number of channels…”.** We thank the reviewer for this helpful suggestion. In the revised manuscript, we have expanded Section 3 to describe in more detail how the intermediate layers reduce and subsequently restore the number of channels, explicitly specifying the squeeze and excitation block and indicating where in the architecture these transformations are applied. Additionally, we reference this expanded description in Section 4.2 as "These layers reduce and then restore the number of channels, controlled by a reduction factor $r \in \mathrm{Z}_+$ as shown in section 3."
>
> **Response regarding dilation rates and adaptive receptive fields.** Thank you for raising this point. In our design, the dilation rates are not adaptive: they are fixed hyperparameters chosen per model (and thus per dataset) and remain constant during training. The adaptive part is the SE block, which performs input-dependent channel mixing. For each input, it applies global average pooling followed by two $1 \times 1$ convolutions with GELU and sigmoid to produce channel-wise gates, allowing the network to weight channels (and thus feature patterns) based on the input. In this sense, this provides explicit input-adaptive feature mixing through channel-wise recalibration mechanisms. Dilated convolutions simply expose larger receptive fields by using different dilation factors along the spatial dimensions. This increases flexibility but is not ``adaptive to the dataset'' in the sense of being learned. We have clarified in the revised manuscript that adaptivity in the paper refers specifically to the SE-based channel-wise recalibration, not to the dilation rates (Section 4.2, first paragraph).
>
> **Response regarding how the time in Table 2 is computed.** In fact, it was not the inference time that was reported, but rather the training time per epoch. The explanation is found under section 6 paragraph 1, in the discussion.
>
> **Response regarding correcting typos (DNCO formatting; equation order in Section 4.1).** We agree with the reviewer and have corrected the typos in the revised manuscript.

---

### Review · Reviewer_kvk2 · 2025-11-10

**Summary Of Contributions:**

This manuscript proposes D-SENO, a neural operator learning model that combines local dilated convolutions with global squeeze-and-excitation blocks, enabling it to combine local and global information much more efficiently than current state of the art transformer or Fourier neural operators. The authors demonstrate the effectiveness of their model on four standard PDE learning benchmarks: airfoil potential flow, Poiseuille pipe flow, heterogeneous Darcy flow, and the time-dependent incompressible Navier–Stokes equations.
show state-of-the-art performance at a fraction of the computational cost.

Strengths:
- The methodology as presented is sound and details of the model are well explained. The contributions are contextualized well within prior relevant work, the framing of the architecture within three stages lifting, processing and projection makes it easy to understand.
- The evaluation across four standard PDE benchmarks clearly demonstrate the effectiveness of the model, with significant gains in efficiency while still preserving state-of-the-art accuracy on all the problems.
- The Appendix also contains a series of ablation studies showing the impact of the different design choices made by the authors. Overall, the results show that the model accuracy consistently improves with depth and is not too sensitive to the exact choice on the dilation pattern.
- The implementation and inclusion of the improved FNO+ model is also a great contribution in itself, the authors should probably explain the improvements in better detail in the Appendix.

Weaknesses:
- No major weaknesses, though a few minor ones that can improve the quality of the paper
- Some theoretical grounding for the approach would be welcome, it is currently not clear why the dilated convolutions and SE-blocks should lead improvements in operator learning and generalization, particularly because the appendix shows minor improvements in accuracy from the SE-blocks, unlike the claim by the authors themselves. This minor accuracy improvement can easily be justified simply the the increased number of parameters, rather than the effective exploitation of global information. Maybe some comparison of a smaller D-SENO model vs a model without SE-blocks of a similar parameter count would result in some more insights?
- The paper mentions “dataset-specific dilation rates,” but doesn’t describe how these are selected. This makes it harder to assess whether the model can easily be tuned and generalize on new datasets.
- The authors mention that the model's purely convolutional nature allows it to be applied on irregularly sampled data, unlike FNOs, but do not really demonstrate this claim.
- No discussion of model limitations, beyond just the brief mention of difficulties with periodic data such as the Navier-Stokes equations problem. It would be nice to see the authors discussing other potential limitations or drawbacks of the current approach.

**Audience:**

Yes

**Audience Explanation:**

The field of ML for physics and PDE learning in particular has been increasingly popular and relevant recently, so I am sure that most TMLR readers working in this or related fields would be interested to learn of this model, given it's good performance and relative computational efficiency.

**Broader Impact Concerns:**

There are no broader impact concerns that I am aware of.

**Claims And Evidence:**

Yes

**Claims Explanation:**

Overall, most of the claims made are supported by accurate, convincing and clear evidence. As mentioned before, the authors evaluate their model on a series of standard PDE benchmarks and demonstrate both state-of-the-art performance and high computational efficiency.
The only claim that I would say is not well supported is the claim that the inclusion of the SE block significantly improves the accuracy of the model, which I feel is not supported by the results in the appendix. However, as I mentioned the authors can perhaps do an additional small experiments to better support their claim.

**Requested Changes:**

None of these changes would be critical for my recommendation, but I will sort in order of decreasing potential to improve the work:

- Provide additional analysis comparing models with and without SE-blocks at matched parameter counts to verify whether performance gains stem from architectural benefits rather than increased model size.
- Describe the procedure for choosing “dataset-specific” dilation rates — whether it’s heuristic, grid-searched, or adaptive — and discuss how this process might generalize to new PDE datasets.
- Either add a small experiment or discussion to support the claim that D-SENO can handle irregular or nonuniformly sampled data (e.g., masked grids or unstructured domains).
- Expand the conclusion or discussion section with a short reflection on the current model’s limitations
- Describe the improvements made to the original FNO model in more detail in the Appendix
- Include a brief theoretical or conceptual justification for why dilated convolutions and SE-blocks should enhance operator learning and generalization, beyond empirical evidence.

---

> ### Author Response · Authors · 2025-12-14
> **Response to Reviewer kvk2**
>
> We thank the reviewer for their thoughtful comments.
>
> **Response regarding comparing models at matched parameter counts.** We thank the reviewer for their suggestion. We have performed more experiments to support our claim. The results are shown in Table 1. These results have also been added to respective datasets results Tables in the appendix in the revised manuscript. To isolate the effect of the SE block from a simple increase in model capacity, we built a control variant in which the SE block is removed and replaced by two additional 1×1 convolutional layers (with a GELU activation) configured to have a comparable number of parameters as the SE-based model. This preserves a similar number of parameters to the original SE configuration while removing the explicit squeeze-and-excitation mechanism. In our experiments, the SE-based model consistently achieves better performance than the parameter-matched baseline on the airfoil, pipe, and Navier–Stokes datasets, while the baseline obtained by simply increasing the number of parameters performs better only on the Darcy flow task. This pattern indicates that the performance gains largely stem from the SE architecture rather than mere model size, and that overall our SE-based approach provides the stronger model across datasets.
>
> **Table 1: Model performance summary on the dataset.**
>
> | Model | # DS Blocks | Time per Epoch (s) | # Parameters (M) | Relative $L_2$ Error |
> |---|---:|---:|---:|---:|
> | Model Airfoil-G w/o SE (PM) | 7 | 2.05 | 1.042 | 0.0056 |
> | Model Pipe-G w/o SE (PM) | 7 | 4.01 | 1.306 | 0.0037 |
> | Model Darcy-F w/o SE (PM) | 6 | 0.99 | 0.505 | 0.0046 |
> | Model NS-A w/o SE (PM) | 8 | 7.53 | 0.666 | 0.1433 |
>
>
> **Response regarding “dataset-specific” dilation rates and their generalization .** The dilation rates in our architecture are treated as dataset-specific hyperparameters rather than learned quantities. In practice, we choose them with a few manual trials, starting from a simple, well-spaced pattern of dilation that spans both local and more global spatial scales. This is a
> heuristic procedure, not an exhaustive grid search or an adaptive method.
> Our ablation study includes, for each model, an alternative configuration
> (”alt”) that uses a different set of dilation rates than the best-performing
> setup. The performance of these ”alt” variants remains very close to the
> best model, indicating that the method is not highly sensitive to the exact
> values, as long as the dilations are reasonably well spread across scales.
> For new PDE datasets, one can simply reuse our default dilation pattern or select a similarly well-spread set of dilation tailored to the spatial resolution of the problem. Our results with the alternative configuration
> suggest that this light-touch, heuristic choice is sufficient, and no complex
> adaptive or large-scale hyper-parameter search is needed. We have added this brief description of selection of dilation rates in the revised manuscript in the appendix in section "A.1.3 Dilation rate impact".
>
> **Response regarding handling irregular or non-uniformly sampled data.** Thank you for pointing this out. To avoid any misunderstanding, we confirm that the current D-SENO formulation is evaluated only on regular grids. We have not tested our model on unstructured data, as we followed commonly adopted benchmark setups for consistent comparison. Evaluating and extending to unstructured domains will be addressed in future work and has been mentioned in current limitations.
>
> **Response regarding reflection on the model’s limitations.** Thank you for this suggestion. We have addressed it by adding a dedicated limitations subsection in the revised manuscript. This new subsection provides a concise reflection on the current model’s key limitations and practical constraints (e.g., inferior performance on Navier-Stokes dataset, extending to unstructured meshes, theoretical analysis and guarantees and manual dilation rates selection), and it outlines directions for future work to mitigate these issues.
>
> **Response regarding describing the improvements to the original FNO.** We have added a subsection in the appendix in the revised manuscript to describe the improvements to the original FNO in greater detail. In particular, we introduce FNO$^{+}$ as a minimally modified yet stronger baseline and explicitly enumerate the architectural changes relative to the standard FNO: (i) replacing ReLU with GELU throughout, (ii) removing batch-normalization layers, and (iii) implementing complex-valued Fourier multiplication directly via a single einsum-based operation. We further clarify that all other components (lifting/projection layers, Fourier block structure, and connectivity) are kept identical to the original FNO. Finally, we further clarify that we report additional experimental variations in the appendix (e.g., different numbers of retained Fourier modes and lifting widths) to assess the impact of Fourier-mode resolution and model capacity.

---

### Review · Reviewer_pPAC · 2025-12-02

**Summary Of Contributions:**

This paper proposes D-SENO, a convolutional neural operator for solving PDEs that combines dilated convolutions with squeeze-and-excitation (SE) blocks. The method is designed to model long-range spatial dependencies while maintaining computational efficiency by avoiding Fourier layers and self-attention.  Experiments on four PDE benchmarks—airfoil flow, pipe Poiseuille flow, Darcy flow, and Navier–Stokes—show that D-SENO achieves competitive or superior accuracy to neural-operator and transformer-based models, and demonstrates ~20× speed improvements in maximum. Ablation studies further assess depth, dilation patterns, SE impact, and resolution robustness.

**Audience:**

No

**Audience Explanation:**

The degree of interest is highly moderate due to limited framework novelty and missing empirical evidence.

**Claims And Evidence:**

No

**Claims Explanation:**

- Limited Novelty

Apart from combining two existing techniques, dilated convolutions and squeeze-and-excitation, this paper has limited novelty. This paper should provide a more comprehensive comparison between D-SENO and existing work (CNO and DCNO), both conceptual and empirical, to demonstrate the key differences and improvements.


- Insufficient Experimental

The most important baselines, CNO and DCNO, are not included in the experiment. Given that they are the most related works, their absence makes it comparison unfair. Although Table 2 presented results for more than ten baselines, it is unclear which results were reproduced by the authors and which numbers were directly copied or from original papers. Also, this paper does not specify hyperparameters for these baselines, making it unclear whether the baselines are well-tuned.

- Reproducibility Issues

The provided code in the supplementary material does not contain the necessary data for reproduction, e.g., `NACA_Cylinder_X.npy`.

- Claims of Efficiency Lack Critical Measurements

The major contribution claim is a gain in training efficiency, but the comparison in model size and memory requirement is missing. This is critical in understanding potential tradeoffs.

**Requested Changes:**

- Add empirical comparisons with CNO and DCNO.
- Specify experiment details about baselines, such as hyperparameters.
- Ensure that the supplementary codebase includes all required data
- Report additional measurements beyond “time per epoch", such as model size comparisons

---

> ### Author Response · Authors · 2025-12-14
> **Response to Reviewer pPAC**
>
> We thank the reviewer for their detailed assessment.
>
> **Response regarding specifying experiment details about baselines, such as hyperparameters.** We appreciate the reviewer’s comment and agree that clarity about experimental settings is important. The hyperparameters and architecture details for our models are reported in Tables 3 and 4 in the appendix, which list the exact configurations used in our experiments. We have made
> this more explicit in the revised manuscript in section 5 under "Baselines and Implementation Protocol" to ensure that readers can easily locate the relevant information.
>
> **Response regarding adding empirical comparisons with CNO and DCNO.** We thank the reviewer for this suggestion. Unfortunately, to the best of our knowledge, there is no publicly available implementation of DCNO, which makes a fair and reproducible empirical comparison infeasible at this time. Regarding CNO, some baselines considered in our study are state-of-the-art (e.g. Transovler) and since our goal is to compare against the strongest available methods on these datasets, we therefore focus on these more competitive baselines rather than including CNO, which is no longer state-of-the-art in this setting.
>
> **Response regarding ensuring the supplementary codebase includes all required data** We thank the reviewer for pointing this out. The datasets used in our experiments are all publicly available. Specifically, the airfoil and pipe flow datasets can be accessed at: https://github.com/neuraloperator/Geo-FNO and the Darcy and Navier–Stokes datasets are available at:
> https://drive.google.com/drive/folders/1UnbQh2WWc6knEHbLn-ZaXrKUZhp7pjt-. We have added these details in the appendix in the revised manuscript in sub-section A.1, first paragraph.
>
> **Response regarding reporting additional measurements beyond “time per epoch”, such as model size comparisons.** We appreciate the reviewer’s suggestion. Parameter counts and model size comparisons are already reported in the ablation study. In
> particular, Tables 5, 7, 9, 11, and 13 include the number of parameters for the respective datasets. We have clarified this more explicitly in sub-section A.1.1 in the revised manuscript to make these comparisons easier to locate.

---

### Decision · Action_Editor_tMJA · 2026-01-19

**Recommendation:** Accept as is

**Audience:**

Yes

**Audience Explanation:**

There are two communities this paper speaks to, first as a baseline for physical system modeling in benchmarking comparisons. Being lightweight, it makes for an excellent comparison. One reviewer specifically called out that it may be interesting for practitioners as well.

**Claims And Evidence:**

Yes

**Claims Explanation:**

This paper looks at the problem of training light surrogate models for physics-driven models. The proposed method is a lightweight operator, controlling the receptive fields with dilated convolutions and the recalibration of feature channels for multiscale processes with squeeze-excite modules.

The reviewers had questions about the reproducibility and the clarity, in particular, on where the gains specifically originate, which have been addressed during the rebuttal. Due to the simplicity of the method, it was pointed out that it may be of interest to practitioners building PDE surrogate architectures.